# Association of Delirium and Depression with Respiratory and Outcome Measures in COVID-19 Inpatients

**DOI:** 10.3390/jpm13081207

**Published:** 2023-07-29

**Authors:** Alessio Simonetti, Cristina Pais, Vezio Savoia, Maria Camilla Cipriani, Matteo Tosato, Delfina Janiri, Evelina Bernardi, Ottavia Marianna Ferrara, Stella Margoni, Georgios D. Kotzalidis, Daniela Chieffo, Massimo Fantoni, Rosa Liperoti, Francesco Landi, Roberto Bernabei, Gabriele Sani

**Affiliations:** 1Menninger Department of Psychiatry and Behavioral Sciences, Baylor College of Medicine, Houston, TX 77030, USA; alessio.simonetti@guest.policlinicogemelli.it; 2Department of Neuroscience, Section of Psychiatry, Fondazione Policlinico Universitario Agostino Gemelli IRCCS, 00168 Rome, Italy; delfina.janiri@uniroma1.it (D.J.); ottaviaferrara@icloud.com (O.M.F.); stella.margoni98@gmail.com (S.M.); gabriele.sani@unicatt.it (G.S.); 3Department of Geriatrics, Fondazione Policlinico Universitario A. Gemelli IRCCS, 00168 Rome, Italymariacamilla.cipriani@policlinicogemelli.it (M.C.C.); matteo.tosato@policlinicogemelli.it (M.T.); rosa.liperoti@unicatt.it (R.L.); francesco.landi@unicatt.it (F.L.); roberto.bernabei@policlinicogemelli.it (R.B.); 4Service of Clinical Psychology, Fondazione Policlinico Universitario Agostino Gemelli IRCCS, 00168 Rome, Italy; vezio.savoia@gmail.com (V.S.); danielapiarosaria.chieffo@policlinicogemelli.it (D.C.); 5Department of Neurology and Psychiatry, Sapienza University of Rome, 00185 Rome, Italy; 6Department of Neuroscience, Mental Health, and Sensory Organs (NESMOS), Faculty of Medicine and Psychology, Sant’Andrea Hospital, Sapienza–Università di Roma, 00189 Rome, Italy; 7Department of Life Sciences and Public Health, Faculty of Medicine and Surgery, Università Cattolica del Sacro Cuore, 00168 Rome, Italy; 8Laboratory and Infectious Diseases Sciences, Fondazione Policlinico Universitario Agostino Gemelli IRCCS, Largo F. Vito 1, 00168 Rome, Italy; massimo.fantoni@unicatt.it; 9Infectious Diseases Section, Department of Safety and Bioethics, Università Cattolica del Sacro Cuore, 00168 Rome, Italy; 10Department of Geriatrics, Università Cattolica del Sacro Cuore, 00168 Rome, Italy; 11Department of Neuroscience, Section of Psychiatry, Università Cattolica del Sacro Cuore, 00168 Rome, Italy

**Keywords:** COVID-19, depression, delirium, hospitalization, prognosis, partial arterial oxygen pressure (PaO_2_), oxygen saturation (SpO_2_), respiratory outcomes

## Abstract

Delirium (DEL) and depression (DEP) may impair the course and severity of acute respiratory illness. The impact of such syndromes on respiratory and outcome parameters in inpatients with COVID-19 needs clarification. To clarify the relationship between DEL and DEP and respiratory outcome measures, we enrolled 100 inpatients from COVID-19 units of the Fondazione Policlinico Universitario Agostino Gemelli IRCCS of Rome. Participants were divided into those with DEL, DEP, or absence of either delirium or depression (CONT). Delirium severity was assessed with the Neelson and Champagne Confusion Scale (NEECHAM). Psychopathology was assessed with the Hamilton Rating Scale for Depression (HAM-D), the Hamilton Rating Scale for Anxiety (HAM-A), and the Brief Psychiatric Rating Scale (BPRS). Dependent variables include: (a) respiratory parameters, i.e., partial pressure of oxygen in arterial blood (PaO_2_), oxygen saturation (SpO_2_), ratio between arterial partial pressure of oxygen (PaO_2_), and fraction of inspired oxygen (PaO_2_/FiO_2_); (b) outcome parameters, i.e., duration of hospitalization and number of pharmacological treatments used during the hospitalization. We investigated between-group differences and the relationships between severity of delirium/depression and the dependent variables. Duration of hospitalization was longer for DEL than for either DEP or CONT and for DEP compared to CONT. NEECHAM and HAM-D scores predicted lower PaO_2_ and PaO_2_/FiO_2_ levels in the DEL and DEP groups, respectively. In DEP, BPRS scores positively correlated with duration of hospitalization. Delirium impacted the course of COVID-19 more severely than depression. The mechanisms by which delirium and depression worsen respiratory parameters differ.

## 1. Introduction

In December 2019, a new acute respiratory disease caused by the severe acute respiratory syndrome coronavirus-2 (SARS-CoV-2) was identified in Wuhan, China [1]. Afterwards, it spread all over the world, causing a pandemic emergency and putting the global healthcare system under extreme pressure [2]. Symptoms of coronavirus disease 2019 (COVID-19) are more commonly mild and do not cause severe respiratory impairment. However, in a considerable proportion of cases, COVID-19 infection might lead to a severe acute respiratory distress syndrome (ARDS), which may be fatal [3]. COVID-19-related SARS is characterized by impairment of respiratory parameters. These include a reduction of the partial pressure of oxygen in arterial blood (PaO_2_), oxygen saturation (SpO_2_), and the ratio between arterial partial pressure of oxygen and fraction of inspired oxygen (PaO_2_/FiO_2_). Furthermore, changes of arterial partial pressure of carbon dioxide (PaCO2), varying from increase to decrease, have been reported [4]. To provide personalized assistance and meet special needs in persons with COVID-19, specialized inpatient units have been implemented in the last two years [5]. A surge or worsening of several neuropsychiatric disorders, such as delirium, dementia, depression, anxiety, phobias, and manic states, was observed in these units [6,7,8,9] with COVID-19, but they are often found in emergency units independently from the pandemic. Among all the aforementioned neuropsychiatric disorders, delirium and depression were the most frequently observed [10].

Delirium is a severe and acute alteration of mental status. Features include loss of time and space orientation, grossly altered thinking and speech, motor agitation, or extreme retardation [11]. Depression is a mental disorder characterized by low mood, anhedonia, loss of drive, disturbed sleep and appetite, and possible suicidal thoughts [12]. Despite substantial agreement in considering delirium and depression as two different entities, some evidence points towards a certain overlap in their clinical picture [13] and underlying pathophysiology [14]. Accordingly, scales frequently related to depression and mental disorders have been used to capture core aspects of delirium, such as sleep and mood alterations [15].

The onset of delirium or depression in inpatients with COVID-19 might affect their prognosis. People in COVID-19 units who developed delirium showed PaO_2_ and PO_2_/FiO_2_ levels shifted towards the pathological side, greater duration of hospitalization, higher rates of in-hospital death, greater use of intensive ventilators, and higher admission rates to intensive care [8,16,17,18,19], although this is not limited to patients with COVID-19, but extends to anyone with respiratory problems. Regarding depression, the vast majority of studies focused on the effect of COVID-19 severity on the development of depression and anxiety, rather than vice versa [20]. Despite this, Saidi et al. [21] reported a correlation between severity of depressive symptoms and both longer duration of hospitalization and lower SpO_2_ levels in COVID-19 inpatients, although the performed analyses failed to provide the direction of the association. Therefore, we need to better investigate the associations between depression on one hand and the course and severity of COVID-19 in inpatient samples. Since currently delirium and depression are considered as separate, nonoverlapping conditions, although they may frequently co-occur along with dementia in elderly patients [22], research comparing their combined impact on respiratory and clinical parameters in COVID-19 and other acute respiratory conditions has been scanty. Even though worse prognosis has been documented in hospitalized patients with delirium as compared to those with depression in non-COVID hospitalized patients [23], more comparative studies on the effect of these two neuropsychiatric syndromes are needed. Delirium can be differentiated from dementia due to its fluctuating nature, but in cases in which it superimposes dementia, the differential diagnostic algorithm becomes more complicated [24].

The aim of this study was to investigate: (i) possible differences in respiratory and outcome parameters of inpatients with COVID-19 between those having delirium or those with depression or those without delirium or depression; (ii) whether severity of delirium or depression is differentially related with respiratory and outcome parameters in the above two groups. To do so, we recruited inpatients with COVID-19 who developed delirium (DEL), those who developed depression (DEP), and those who did not develop either delirium or depression (CONT). PaO_2_, PaCO_2_, SpO_2_, and PaO_2_/FiO_2_ were used as respiratory parameters, whereas duration of hospitalization and number of drugs used during hospitalization were used as outcome parameters. Since hospitalized patients with delirium have shown a poorer prognosis than those affected by depression or those without either delirium or depression, we expected to find lower PaO_2_, SpO_2_, and PaO_2_/FiO_2_ and higher PaCO_2_ in DEL as compared with DEP and CONT. To investigate the association of severity of delirium/depression with the levels of respiratory parameters and prognosis, we used psychometric scales usually employed to assess delirium, depression, and other psychopathology associated with depression, such as anxiety, manic and psychotic symptoms, irritability, and apathy. We expected that in both in DEL and DEP, severity of either delirium and depression would correlate with worse respiratory and outcome parameters (lower PaO_2_, SpO_2_, PaO_2_/FiO_2_, and higher PaCO_2_, longer duration of hospitalization, and higher number of drugs used during hospitalization). Furthermore, we additionally investigated the association of general psychopathology on functional and outcome measures in DEL. We expected that severity of psychopathology in DEL would positively correlate with impairment of respiratory parameters and worse outcome.

## 2. Materials and Methods

### 2.1. Study Design

This was a cross-sectional study investigating associations between respiratory/outcome measures and psychopathological parameters probing delirium and general psychopathology, anxiety, and depression in patients with COVID-19 infection. Hence, no causal relationships can be established. However, with respect to other studies, which studied through statistical regression the effect of worse respiratory measures on psychopathology, here we used regressions to investigate psychopathological measures as predictors of respiratory function. Study design is shown in Figure 1. Patients were assessed by expert clinicians.

### 2.2. Sample

Inpatients enrolled were consecutively hospitalized for SARS-CoV-2 infection at “Policlinico Universitario Agostino Gemelli IRCCS, Presidio Columbus” of Rome, Italy. The present sample’s enrollment phase started on 14 April 2020 and ended on 20 October 2021. Inclusion criteria were: age > 18 years old, diagnosis of COVID-19, absence of mental retardation, capability of providing informed consent, absence of dementia or other severe neurological disorders. The diagnosis of COVID-19 was determined in accordance with the World Health Organization interim guidance (WHO, 2020, https://www.who.int/publications/i/item/diagnostic-testing-for-sars-cov-2, accessed on 27 July 2023). Informed consent was obtained by participants before entering the study. Patients’ demographic and clinical characteristics (i.e., age, sex, education, marital status, number of comorbidities, number of drug treatments before the admission, and employment) were extracted from each participant’s electronic health records and registered into an electronic database available only to the authors.

Within one week following admission, participants underwent psychopathological assessment including: the Neelon and Champagne Confusion Scale (NEECHAM) [25], which assesses the level of delirium; the Hamilton Rating Scale for Depression (HAM-D) [26], which rates the severity of depressive symptoms; the Hamilton Rating Scale for Anxiety (HAM-A) [27], which measures the severity of anxiety symptoms, the Young Mania Rating Scale (YMRS) [28], which assesses the severity of manic symptoms, and the 18-item Brief Psychiatric Rating Scale (BPRS) [29], which rates the severity of psychopathology. Such assessment was performed by a psychologist who was trained through telemedicine. Laboratory tests regarding respiratory parameters relevant to this study were collected within 12 h from the psychopathological assessment.

We divided the selected sample in three groups. Patients having a NEECHAM confusion scale ≤24 were labeled as patients with delirium (DEL). Those having a NEECHAM confusion scale >24 and a HAM-D score of >7 were labeled as patients with depressive symptoms (DEP). We chose the NEECHAM instead of the Confusion Assessment Method (CAM) [30] to evaluate delirium because the former showed better sensitivity in detecting delirium cases [31]. Those scoring >24 on the NEECHAM confusion scale and ≤7 on the HAM-D, ≤17 on the HAM-A, and ≤11 on the YMRS were considered with absence of delirium and psychopathology and included as controls (CONT). Then, the stability of the present diagnoses was tested during the next days as follows: (i) participants with depressive symptoms lasting less than the two minimum days required to define a brief depression [32] and those having a delirium duration of less than the average five described in a recent observational study on delirium in COVID-19 inpatients [18] were excluded from the study; the five-day cutoff was chosen to enhance comparability of our study with most studies in literature; (ii) enrolled individuals belonging to the DEL group who showed resolution of delirium and later developed depressive symptoms and patients belonging to the DEP group who later developed delirium were excluded from the study; (iii) participants who developed coma, required intubation, and were transferred to the intensive care unit or died were excluded from the study. We conducted between-group analyses to assess for possible relationships between severity of delirium and/or depression and respiratory function.

### 2.3. Assessment Tools

The *NEECHAM confusion scale* [25] was developed to assess delirium. The NEECHAM Confusion Scale evaluates components of cognitive status (attention, alertness, verbal and motor response, memory, and orientation), observed behavior and performance ability (general appearance and posture, sensory-motor performance, and verbal responses), and vital function (vital signs). The scores may range from 0 (minimal function) to 30 (normal function); the cut-off point is 24. The range from 0–24 points indicates the presence of delirium; specifically, 30–27 is patient non-delirious, 26–25 is at risk, 24–20 is early to mild confusion, and 19–0 is moderate to severe confusion.

The *HAM-D* is a clinician-rated scale assessing the severity of depressive symptoms. It has various versions, with short and long versions (17-, 21-, 24-, 28-, and 29-item), but for the final rating, only the first 17 items count. Seven items are scored 0–2, one 0–3, and each of the other nine is scored 0–4 points. The possible range is 0–53, with scores of 0–7 being considered as normal or in remission, 8–16 suggest mild depression, 17–23 moderate depression, and >24 indicate severe depression [26].

The *HAM-A* measures the severity of anxiety symptoms. The scale consists of 14 items, each referring to a series of symptoms. It measures both psychic (mental agitation and psychological distress) and somatic anxiety (physical complaints related to anxiety). Each item is scored on a scale of 0 (not present) to 4 (severe), with a total score range of 0–56, where <17 indicates mild severity, 18–24 mild to moderate severity, 25–30 moderate-to-severe, and >30 severe anxiety [27].

The *YMRS* is a clinician-administered tool used to rate the severity of manic symptoms. It is an 11-item scale assessing mood, motor activity/energy levels, interest in sex, sleep, irritability, rate and frequency of speech, flight of ideas, grandiosity, aggressive behavior, appearance, and insight into current presentation. Seven items are rated 0–4 and four 0–8, for a range of 0–60. Total scores ≤12 indicate no mania or remission, 13–19 minimal manic symptoms, 20–25 mild mania, 26–37 moderate, and 38–60 severe mania [28].

The *BPRS.* The original version is an 18-item rating scale designed to rate general psychopathology with particular attention to psychosis. Each of the 18 items is designed to represent a discrete symptom area. Five of the items (tension, emotional withdrawal, mannerisms and posturing, motor retardation, and uncooperativeness) are based on observations of the patient. The remaining 13 items are based on the patient’s verbal report. Items are rated on a 7-point Likert scale, from 1 “not present” to 7 “extremely severe”, with scores ranging from 18 to 126 (achieved through summing the item scores). There are no cutoffs, with higher scores reflecting more severe psychopathology [29].

### 2.4. Respiratory Parameters

Respiratory parameters were collected within 12 h from the psychopathological evaluation. Respiratory parameters taken into account were the following: PaO_2_, PaCO_2_, SpO_2_, PaO_2_/FiO_2_.

SpO_2_ was assessed by peripheral oxygen saturation using the Nonin Avant 9600 Tabletop Oximeter for pulse oximetry, with an error range of 2–3% (FDA approved). PaO_2_ and PaCO_2_ were analyzed from an arterial blood gas analysis performed by Columbus physicians who were blind as to the patient’s participant status. In detail, blood samples were obtained from the patients’ radial arteries and then tested with a highly accurate Abbott i-Stat 300 handheld automated blood analyzer. Since PaO_2_/FiO_2_ showed greater reliability than PaO_2_ alone in defining COVID-19-related acute respiratory distress syndrome, [4] this parameter was additionally calculated.

### 2.5. Outcome Parameters

We chose as outcome parameters duration of hospitalization (in days) and number of drug treatments (total number of drugs received simultaneously on admission and while hospitalized).

### 2.6. Other Scales Related to the Clinical Status

The Clinical Frailty Scale (CFS) [33], the Activities of Daily Living scale (ADL) [34], and the Instrumental Activities of Daily Living scale (IADL) [35] were administered on admission. ADL and IADL assess global functioning. Both refer to daily activities. The first is targeted to taking care of one’s own body, the second to more complex activities such as financial management and housekeeping. Higher scores correspond to higher levels of autonomy. For all used scales, we used their validated Italian counterparts.

### 2.7. Ethics

The protocol of the present study was approved by the ethical committee of the “Policlinico Universitario Agostino Gemelli IRCCS” (ID: Prot. Code 0017446/20) and conducted according to the Fondazione Policlinico Universitario Agostino Gemelli Ethics Committee guidelines and in accordance with the Principles of Human Rights, as adopted by the World Medical Association at the 18th WMA General Assembly, Helsinki, Finland, June 1964 and subsequently amended at the 64th WMA General Assembly, Fortaleza, Brazil, October 2013. All participants gave their written informed consent to participate in the study after having received a complete explanation of study procedures.

### 2.8. Statistical Analyses

#### Demographic and Clinical Characteristics

Between-group differences in demographical and clinical characteristics were investigated using multiple one-way analyses of variance (ANOVA) and *chi*-square (*χ*^2^) tests. ANOVA was used for continuous variables. In each ANOVA, the three groups (DEL, DEP, CONT) were used as independent variables; continuous variables of interest were used as dependent variables. Tukey HSD was used as a post hoc test. The *χ*^2^ test was used for categorical variables.

### 2.9. Respiratory and Outcome Parameters

Between-group differences in respiratory and outcome parameters (i.e., PaO_2_, PaCO_2_, SpO_2_, PaO_2_/FiO_2_ levels, duration of hospitalization, number of pharmacological treatments used) were investigated using multiple one-way analyses of variances (ANOVA). In each ANOVA, the three groups (DEL, DEP, CONT) were independent variables, whereas respiratory and outcome parameters were dependent variables. Results were Bonferroni-corrected. To avoid type-one errors, ANOVA analyses were preceded by a multivariate analysis of variance (MANOVA). In the MANOVA, groups (DEL, DEP, CONT) were the independent variables, whereas respiratory and outcome parameters were the dependent variables.

### 2.10. Correlation between Severity of Delirium, Depression, and Associated Psychopathology on Respiratory and Outcome Parameters

Correlation between severity of delirium, depression, and associated psychopathology with levels of respiratory and outcome parameters in DEL and DEP were investigated with multiple linear regressions on the entire group of DEP and DEL. First, linear regressions were run using HAM-D, HAM-A, YMRS, and BPRS total scores as predictors, whereas respiratory and other clinical outcome parameters (i.e., PaO_2_, PaCO_2_, PaO_2_/FiO_2_, SpO_2_ levels, days of hospitalization, and number of drug treatments used during hospital stay) were outcome variables. Interactions between predictors and DEL + DEP group on the outcome variables were also investigated. When significant interactions were found, linear regressions were performed for the DEL and DEP groups separately. Second, only in DEL, linear regressions were performed between NEECHAM scale scores and functional and outcome parameters. Spearman’s *ρ* correlations were performed to investigate the direction of correlations between variables.

### 2.11. Effect of Possible Confounding Variables

Possible effects of confounding variables were also investigated. Results of one-way ANOVA analyses were corrected for the effect of demographic and clinical variables showing significant differences among groups. Therefore, multiple one-way analyses of covariance (ANCOVAs) were run. In each ANCOVA, the three groups (DEL, DEP, CONT) were used as independent variables; continuous variables of interest were used as dependent variables and possible confounding variables were used as covariates. The effects of possible confounding variables on predictors of respiratory and other clinical outcome parameters were investigated with multiple linear regressions. In each regression, demographic (i.e., age, sex, employment status, living conditions, and education) and clinical variables (i.e., number of comorbidities, BMI, presence of pneumonia, presence of dyspnea, presence of hyperthermia, presence of cough, number of drug treatments prior to admission, CFS, ADL, and IADL scores) were predictors, whereas PaO_2_, PaCO_2_, SpO_2_, PaO_2_/FiO_2_, days of hospitalization, and number of drugs used during hospitalization were outcome variables. We entered all significant demographic and clinical predictors in a multivariate model, along with significant predictors among the scales (NEEHAM, HAM-D, HAM-A, YMRS, and BPRS). Separate multivariate models were performed for each dependent variable (PaO_2_, PaCO_2_, SpO_2_, PaO_2_/FiO_2_, days of hospitalization, and numbers of drugs used during hospitalization) for DEL and DEP groups separately. To ensure adequate power, we calculated the needed sample size through the ClinCalc sample size calculator; to ensure a power of 0.8, a sample of 116 patients would have been adequate (two groups of 58 each).

## 3. Results

### 3.1. Sample

Of the initially screened 136 patients, 100 inpatients were included in the study.

### 3.2. Demographic and Clinical Characteristics 

Results are shown in Table 1. Groups differed for age, education, number of comorbidities, number of drug treatments before hospitalization, and levels of ADL and IADL before admission. Gender differences only approached significance. Post hoc analyses showed that DEL were older, had fewer years of education, more comorbidities, and assumption of drug treatment before admission, and displayed lower scores of IADL than both DEP and CONT. DEL also scored lower on ADL than CONT. No differences were found between DEP and CONT.

### 3.3. Psychopathological Scales

Results are shown in Table 1. Significant differences among groups were found in HAM-D, HAM-A, and BPRS scores. Post hoc analyses revealed that DEP scored higher on HAM-D, HAM-A, and BPRS as compared to both DEL and CONT, even though differences with DEL only approached significance. On the other hand, DEL scored higher on NEECHAM than both DEP and CONT, and scored higher on HAM-D, HAM-A, and BPRS than CONT.

### 3.4. Respiratory and Outcome Parameters

The preliminary MANOVA found a significant effect of group (Wilks’ *lambda* = 0.51; Degrees of Freedom (DF) = 12, F = 5.64; *p* < 0.01). One-way ANOVA revealed differences in SpO_2_, duration of hospitalization, and number of drug treatments during hospitalization. Differences in PaO_2_ levels only reached a trend-level significance. Post hoc analyses revealed that DEL had lower levels of SpO_2_ and assumed a greater number of drug treatments during hospitalization than both DEP and CONT, even though differences between DEL and CONT regarding the levels of SpO_2_ only reached a trend. As regards duration of hospitalization, DEL were hospitalized for longer periods than both DEP and CONT, whereas DEP were hospitalized for more days than CONT. These results are presented in Table 2.

### 3.5. Correlation between Severity of Delirium, Depression, and Associated Psychopathology with Respiratory and Outcome Parameters

Linear regressions showed no effect of psychopathology on functional and outcome parameters of the whole sample (DEL + DEP). HAM-D-by-group and HAM-A-by-group interactions were found for PaO_2_ (F = 6.34; *p* = 0.01; F = 4.56 *p* = 0.04, respectively), and PaO_2_/FiO_2_ levels (F = 9.42; *p* < 0.01; F = 6.83; *p* = 0.01, respectively). A BPRS-by-group interaction was found for PaO_2_/FiO_2_ levels and duration of hospitalization (F = 8.69; *p* < 0.01). Linear regressions showed that in DEP, HAM-D total scores predicted PaO_2_ and PaO_2_/FiO_2_ levels. HAM-A total scores predicted PaO_2_/FiO_2_ levels, while BPRS total scores predicted duration of hospitalization. Spearman’s correlation analyses showed that HAM-D total scores negatively correlated with PaO_2_ and PaO_2_/FiO_2_ levels; HAM-A negatively correlated with PaO_2_/FiO_2_ levels, and BPRS positively correlated with duration of hospitalization. In DEL, no correlation with psychopathology was found.

In DEL, a significant effect of the NEECHAM Confusion Scale Scores on PaO_2_ and PaO_2_/FiO_2_ was found. PaO_2_ and PaO_2_/FiO_2_ levels negatively correlated with levels of confusion. Results are presented in Table 3 and Appendix A.

### 3.6. Effect of Possible Confounding Variables

Age, education, number of comorbidities, number of pharmacological treatments before admission, and ADL and IADL scores before admission entered each ANCOVA. After adjusting for these variables, differences regarding number of drug treatments used during hospitalization switched to not significant, whereas differences regarding PaO_2_/FiO_2_ became significant (F = 3.92; *p* = 0.02). Hence, post hoc analyses were performed to investigate between-group differences regarding PaO_2_/FiO_2_. These showed that DEL have a lower PaO_2_/FiO_2_ than DEP. Other among-group differences did not change. As regards the effect of possible confounding variables on predictors of respiratory and outcome parameter alterations, multiple linear regression was performed only in the DEP and DEL groups. In DEP, drug treatments prior to admission and presence of dyspnea significantly predicted PaO_2_ and PaO_2_/FiO_2_ levels. Being employed, ADL and IADL total scores predicted the length of hospitalization. In DEL, presence of dyspnea predicted PaO_2_/FiO_2_ levels, whereas presence of fever predicted both PaO_2_ and PaO_2_/FiO_2_ levels.

The multivariate regression analysis in DEP confirmed the predictive value of HAM-D on PaO_2_ and PaO_2_/FiO_2_, while HAM-A became not significant. Since ADL and IADL were highly correlated, only ADL was retained in the multivariate model regarding the duration of hospitalization. After accounting for the effect of the confounding variables, the effect of BPRS on duration of hospitalization remained significant. In DEL, multivariate regressions confirmed the predictive value of NEECHAM confusion scale on PaO_2_ and PaO_2_/FiO_2_ levels. Results are presented in Table 3 and Appendix A.

## 4. Discussion

Results can be summarized as follows: DEL showed lower levels of SpO_2_ and PaO_2_/FiO_2_ than DEP. DEL showed longer duration of hospitalization than either DEP and CONT, whereas DEP showed intermediate length of stay between DEL and CONT. Lower levels of PaO_2_ and PaO_2_/FiO_2_ were predicted by greater delirium severity in DEL and greater depression severity in DEP. Additionally, in DEL, greater levels of general psychopathology, as assessed through the BPRS, predicted longer duration of hospitalization.

Findings of longer duration of disease in DEP, as compared with CONT, overlap with those already present in the literature [17] and confirm the poorer prognosis of the occurrence of delirium in the context of severe, acute disease. Differently from what expected, respiratory parameters between DEL and CONT did not differ. Such a finding is in contrast with those of Ticinesi et al. [8] and Khan et al. [18] who documented poorer PaO_2_/FiO_2_ levels in patients with delirium, as compared with those without. Differences in study designs might account for these discrepancies. Ticinesi et al. [8] measured respiratory parameters on admission and before the onset of delirium, whereas Khan et al. [18] did not mention the timeline of vital sign measurements and delirium onset. Since low oxygen levels might cause delirium [36], the differences found could be a proxy of differences in blood oxygen levels on admission. Furthermore, in both previously mentioned studies, presence of depression or related psychopathology were not investigated. Since presence of delirium showed poorer prognosis than presence of depression in acute inpatient units [23], it might be possible that differences found in the other two studies might be driven by underlying psychiatric disorders.

On the other hand, between-group differences on duration of hospitalization suggest that depression impacts the course of COVID-19, even though the impact is less pronounced than the one produced by delirium. Absence of direct comparisons between patients with delirium and depression in COVID-19 inpatients dampens the formulation of possible hypotheses on why DEL showed greater duration of hospitalization than DEP. In non-COVID-19 contexts, depression and delirium did not affect hospitalization rates or duration [37]. On the other hand, results are aligned to those documenting higher rates of functional decline, nursing home admission, and death in people with delirium as compared with people with depression in acute non-COVID-19 inpatient units [23,38,39]. Therefore, it could be argued that functional impairment brought by delirium is greater in magnitude than the one seen in depression.

Regression analyses showed that in DEP, duration of hospitalization was predicted by severity of psychopathology whereas in DEL, it was not. The present findings corroborate hypotheses on the different nature of depression and delirium [13,40] and highlight that delirium and depression influence the duration of hospitalization through different mechanisms. Delirium could increase hospital stay through mechanisms that do not involve psychopathology. Patients with delirium are more likely to fall, and less likely to move effectively and to maintain adequate hydration and nutrition and be less compliant with healthcare interventions [41]. In case of lack of arousal, there might be complications, such as aspiration pneumonia and acute kidney injury, that can further spiral a downwards trajectory and worsen delirium [11].

In DEP, greater severity of general psychopathology, as measured with the BPRS, was associated with longer duration of hospitalization. This finding is in contrast with our hypothesis of an effect of depressive symptoms on this variable. The BPRS investigates a broader range of behavior, which might be also present in depression and cannot be captured by classical mood scales [42,43], such as dysphoria, psychotic features, and apathy. These features might influence duration of hospitalization in different ways. Paranoid delusions might lead to nonadherence to drug treatment, possibly due to the belief of being poisoned. On the other hand, depressive delusions might manifest as ideas of being dead, with the patient being likely to quit any adherence to treatment and rehabilitation programs [44,45]. Apathy can lead to nonadherence with treatment and rehabilitation due to lack of motivation [46], whereas dysphoria can induce oppositional behavior [47,48]. All these factors have been related to increased length of stay in psychiatric and non-psychiatric units [49,50] and might have played a role in our results.

In DEL, greater severity of delirium predicted lower levels of PaO_2_ and PaO_2_/FiO_2_. Severity of delirium has been mainly investigated as a consequence, rather than a cause, of reduced oxygenation in inpatients with COVID-19 [51,52,53] or other pulmonary disorders [54] and a direct effect of severity of delirium on PaO_2_ and PaO_2_/FiO_2_ levels has not being investigated. Nevertheless, results are in line with studies documenting an increase of use or failure of ventilatory support devices commonly used in COVID-19-associated acute respiratory distress syndrome (ARDS), such as noninvasive ventilation (NIV) [55,56]. The mechanisms by which delirium can decrease PaO_2_ and PaO_2_/FiO_2_ may be several. Individuals with delirium might be unable to remove excessive airway secretions, could be nonadherent with or have difficulty adapting to the correct use of respiratory devices, or might have an increased risk of aspiration [57]. Interestingly, PaO_2_ and PaO_2_/FiO_2_ reductions in DEP were predicted by severity of depression. The relationship between mood symptoms and respiration has been extensively studied [58,59,60,61], even though the direction of this relationship is still unclear. Even in this case, the correlation between severity of depressive symptoms and PaO_2_ and PaO_2_/FiO_2_ parameters has not been investigated previously. However, our results are aligned with those documenting a relationship between severity of depression and symptoms commonly present during COVID-19, such as chest constriction, night breathlessness, bronchial responsiveness, and dyspnea, [58,62,63] or laboratory test alterations, such as poor forced expiratory volume in 1 s (FEV_1_) [64,65]. We generally found that stronger correlations at univariate analyses survived multivariate analyses, and this regarded depression levels, dyspnea, and number of drugs taken by the patient in the DEP group for both PaO_2_ and PaO_2_/FiO_2_ (Table 3), while in DEL NEECHAM values were similarly correlated with PaO_2_ and PaO_2_/FiO_2_ but not dyspnea, which correlated only with PaO_2_ on multivariate analysis. Given the limited number of variables included in the present analyses, hypotheses on how depressive symptoms might influence PaO_2_ and PaO_2_/FiO_2_ may be only speculative. Since individuals with depression have shown higher rates of chronic lung obstructive disorders, like bronchitis [66,67], poor self-care, missing follow-up visits for coexisting chronic disorders, like chronic obstructive pulmonary disease (COPD) [68,69], the levels of depression might underlie poorer levels of self-care and greater severity of pulmonary obstruction. This might translate to greater reduction of PaO_2_/FiO_2_ levels once COVID-19 is superimposed on this chronic lung disease [70]. Alternatively, this relationship might derive from an additive effect of inflammatory processes. Excessive proinflammatory cytokine production has been recognized as a crucial process in severe COVID-19 and has also been related to the pathophysiology of depression [64,71]. Specifically, IL-6 has been found to be particularly important in the pathogenesis of COVID-19, since it is positively correlated with disease stages and radiological changes [72,73,74]. Higher levels of IL-6 correlated with poorer PaO_2_ levels [3], and with greater severity of depression [75]. Therefore, we might speculate that greater levels of depression could result in greater proinflammatory cytokine production. This proinflammatory state might contribute to the higher level of pulmonary impairment and lower level of PaO_2_ and PaO_2_/FiO_2_ once COVID-19 is established. To further disentangle open questions and reduce speculations on the nature of these relationships, additional studies are required.

Another important factor that could affect both the expression of delirium and the occurrence of depression is advancing age. In fact, it has been shown that age is the most important factor determining the occurrence of delirium in surgery wards [76], and it could be so in ICUs [77], where all too often patients with severe COVID-19 are referred [78]; in all cases, higher age increases the odds of people developing delirium and worsens outcomes. Although late-life depression has its own clinical peculiarities [79], the relationship of the onset of depression across ages is still developing and may be complex [80]. It appears that delirium and depression share common pathophysiological mechanisms and can trigger one another [13]. Their interrelations with age deserve to be better studied in future work.

In our study, patients with delirium were taking a higher number of medications on admission. This is not unexpected, as the number of medications proved to be predictive of future delirium [81,82,83,84], but also patients with depression are likely to be treated with anticholinergic drugs, which may induce confusion and delirium [85]. However, in our sample, the depressive group did not differ from nondepressive-nondelirium controls.

### Limitations

We should mention several caveats regarding our study. First, the relatively small sample size limits the generalizability of results. Second, the lack of a longitudinal design prevents us from exhaustively clarifying the dynamics between delirium/depression and respiratory and outcome variables. In particular, it is not possible to assess the presence of PaO_2_ or PaO_2_/FiO_2_ reduction before the onset of symptoms of delirium/depression. This limits the definition of a causal relationship between neuropsychiatric disorders and respiratory and outcome parameters. Third, a more comprehensive evaluation of other variables involved in the relationship between neuropsychiatric symptoms and respiratory and outcome variables are warranted to reduce the speculations on possible relationships and between-group differences. Furthermore, the analysis of delirium is limited to the NEECHAM scale. Fourth, we excluded a wide range of patients (36 out of 136, 26.47%), which cut out also the critically ill patients, thus reducing the generalizability of our results. Fifth, the cohort we used was entirely COVID-19-positive, and we had no non-COVID-19 control group to compare with; hence, we cannot assess whether the differences found were specific to COVID-19. We did not stratify our sample according to age; that is a definite and important factor associated with both delirium and depression, but we entered age as a possible confounder in our analyses of variance and the subsequent regressions. More exhaustive analyses of the depressive dimension and a more comprehensive analysis of delirium might better describe the possible neuropsychiatric areas involved in the aforementioned relationship. Another issue was that we measured respiratory parameters within 12 h after performing psychopathological assessment; 12 h is a long interval that may contribute to interpatient variability; however, most patients had their respiratory parameters assessed soon after psychopathological evaluation. Moreover, we did not examine the relationships between respiratory parameters and individual factors of the Hamilton Depression Rating Scale. This scale has undergone multiple factorializations in the literature, yielding from two to eight factors with no constancy as to which items load on each factor [86]. Hence, we avoided splitting what is better to keep as a whole. Finally, there was no non-COVID-19 control group to compare our results, but including one would require a retrospective group that would also subtract validity from our results. A parallel group was unfeasible in our service. More exhaustive analyses of psychopathology related to the depressive dimension [47,87,88] and more comprehensive analyses of delirium and assessment of other variables involved in the relationship between these two dimensions and respiratory/outcome variables might better disentangle the possible relationship between these two domains.

## 5. Conclusions

The presence of delirium is associated with lower levels of respiratory parameters and poorer prognosis than the presence of depression in COVID-19 units. Furthermore, severity of psychopathology and severity of delirium predict lower levels of PaO_2_ and PaO_2_/FiO_2_. Knowledge brought forth by this study might foster new efforts in early intervention strategies aimed to resolve neuropsychiatric comorbidity in people affected by COVID-19. Resolution of these neuropsychiatric syndromes might improve respiratory symptoms and, consequently, might help enforcing treatment strategies against COVID-19. Further studies will hopefully clarify the impact of delirium and depression on COVID-19 severity and course.

## Figures and Tables

**Figure 1 jpm-13-01207-f001:**
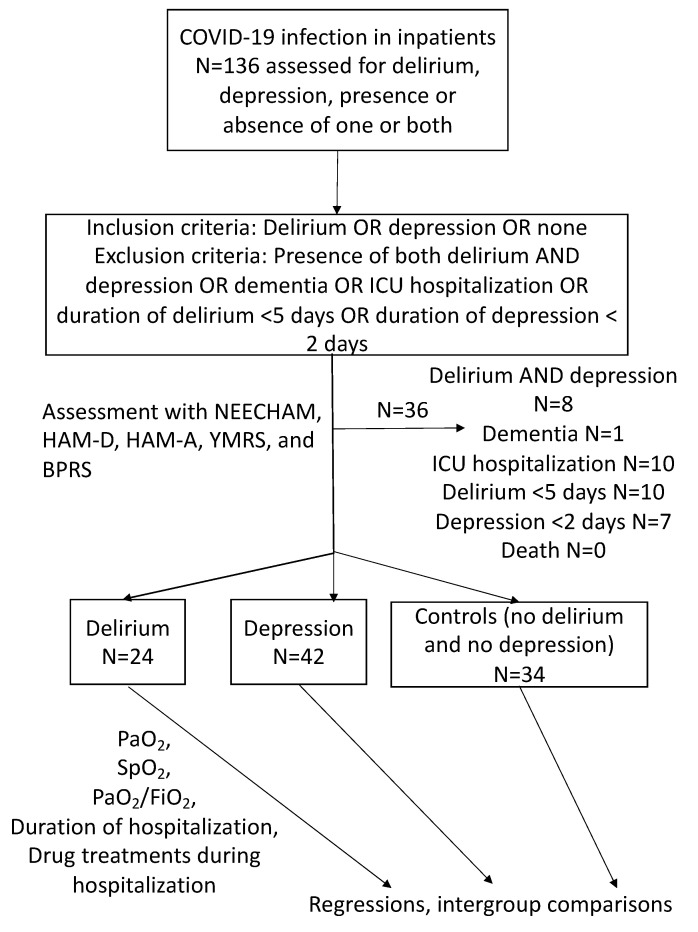
Summary of the inclusion and statistical data processing methods of our study.

**Table 1 jpm-13-01207-t001:** Demographic and clinical characteristics of patients with delirium (DEL), depressive symptoms (DEP), and patients without delirium and psychopathology (CONT).

		DEL (N = 24)	DEP (N = 42)	CONT (N = 34)	*F* or χ^2^	*p*-Value	Post Hoc					
							*DEL* vs. *CONT*		*DEL* vs. *DEP*		*DEP* vs. *CONT*	
Demographics							*p*	*d*	*p*	*d*	*p*	*d*
	Age (y), mean ± SD	74.48 ± 1.22	59.75 ± 1.45	56.58 ± 1.52	12.23	**<0.001**	**<0.001**	12.98	**<0.001**	10.99	0.599	2.13
	Female, n (%)	15 (62.5)	22 (52.4)	27 (79.4.0)	5.99	0.05						
	Employed, n (%)	10 (41.7)	23 (54.8)	21 (61.8)	2.31	0.316						
	Living alone, n (%)	14 (66.7)	28 (66.7)	25 (73.5)	0.48	0.784						
	Education (y), mean ± SD	9.10 ± 3.84	13.17 ± 4.40	12.55 ± 3.93	6.80	**0.002**	**0.014**	0.98	**0.001**	0.98	0.811	0.14
Clinical												
	Comorbidities, mean ± SD	3.58 ± 2.75	1.52 ± 1.76	1.59 ± 1.91	8.68	**<0.001**	**0.001**	0.87	**0.001**	0.91	0.990	0.04
	BMI, mean ± SD	24.83 ± 4.16	26.93 ± 5.29	26.64 ± 4.65	1.30	0.278						
	Pneumonia, n (%)	19 (79.2)	32 (80.0)	27 (79.4)	0.01	0.996						
	Dyspnea, n (%)	10 (41.7)	12 (40.0)	15 (44.1)	2.00	0.368						
	Fever, n (%)	13 (54.2)	24 (60.0)	19 (55.9)	0.24	0.886						
	Cough, n (%)	8 (33.3)	12 (30.0)	11 (32.4)	0.09	0.956						
	Drug treatments prior to admission (n), mean ± SD	4.46 ± 2.64	2.48 ± 2.41	2.03 ± 2.70	6.87	**0.002**	**0.002**	0.78	**0.009**	0.79	0.732	0.18
	CFS before admission, mean ± SD	3.77 ± 2.27	2.77 ± 1.86	3.09 ± 2.09	1.71	0.187						
	ADL before admission, mean ± (SD)	4.09 ± 2.33	4.90 ± 1.97	5.53 ± 1.32	3.85	**0.025**	**0.018**	0.76	0.231	0.38	0.333	0.38
	IADL before admission, mean ± (SD)	4.59 ± 3.39	6.48 ± 2.63	7.41 ± 1.66	7.94	**0.001**	**<0.001**	1.12	**0.017**	0.72	0.275	0.42
Psychopathological												
	NEECHAM	19.58 ± 4.30	29.35 ± 1.51	28.00 ± 5.30	52.291	**<0.001**	**<0.001**	1.74	**<0.001**	0.35	0.311	1.74
	HAM-D, mean ± SD	10.75 ± 5.86	15.44 ± 4.96	4.10 ± 1.94	54.47	**<0.001**	**<0.001**	1.52	**<0.001**	3.01	**<0.001**	1.52
	HAM-A, mean ± SD	9.71 ± 5.46	14.51 ± 5.65	4.00 ± 2.36	41.49	**<0.001**	**<0.001**	1.36	**0.001**	0.86	**<0.001**	2.43
	YMRS, mean ± SD	2.65 ± 1.77	3.50 ± 2.38	2.37 ± 2.46	2.25	0.111						
	BPRS, mean ± SD	32.67 ± 5.20	35.38 ± 4.43	26.23 ± 5.41	30.38	**<0.001**	**<0.001**	1.22	0.086	0.56	**<0.001**	1.85

*Note:* Significant *p*-values (*p* < 0.05) are indicated in **bold**. DEL: people with delirium; DEP: people with depressive symptoms; CONT: controls without delirium or psychopathology. ADL: activities of daily living scale; BMI, body mass index; BPRS, Brief Psychiatric Rating Scale, CFS; Clinical Frailty Scale; *d*, Cohen’s *d*, effect size (≈0.2 small, ≈0.5 medium, and ≈0.8 large); HAM-A, Hamilton Anxiety Rating Scale; HAM-D, Hamilton Depression Rating Scale, IADL, Instrumental Activities of Daily Living scale; n, number; NEECHAM, Neelson and Champagne Delirium Scale; SD, standard deviation; YMRS, Young Mania Rating Scale.

**Table 2 jpm-13-01207-t002:** Respiratory and outcome parameters of patients with delirium (DEL), depressive symptoms (DEP), and patients without delirium and psychopathology (CONT).

		DEL (N = 24)	DEP (N = 42)	CONT (N = 34)	*F* or χ^2^	*p*-Value	Post Hoc					
							*DEL* vs. *CONT*		*DEL* vs. *DEP*		*DEP* vs. *CONT*	
Functional and outcome parameters							*p*	*d*	*p*	*d*	*p*	*d*
	PaO_2_ (mmHg), mean ± SD	70.55 ± 16.03	79.11 ± 12.44	76.94 ± 13.38	3.01	0.053						
	PaCO_2_ (mmHg), mean ± SD	36.33 ± 4.43	36.30 ± 5.09	38.00 ± 5.94	1.07	0.346						
	SpO_2_, (%), mean ± SD	94.38 ± 3.36	96.74 ± 1.75	95.86 ± 2.00	7.89	**0.001**	0.058	0.54	**<0.001**	0.91	0.268	0.53
	PaO_2_/FiO_2_, mmHg mean ± SD	302.25 ± 94.87	354.39 ±71.92	342.85 ± 109.77	2.51	**0.086 ^#^**	0.197	0.40	**0.009**	0.62	0.114	0.12
	Duration of hospitalization (days), mean ± SD	38.54 ± 21.35	20.55 ± 11.34	11.50 ± 6.63	29.313	**<0.001**	**<0.001**	1.71	**<0.001**	1.05	**0.011**	0.97
	Drug treatments during hospitalization (n), mean ± SD	6.67 ± 3.33	4.20 ± 2.69	4.35 ± 2.79	0.6.50	0.002 *	**0.007**	0.76	**0.003**	0.82	0.994	0.05

Note: Significant *p*-values (*p* < 0.05) are indicated in **bold**. DEL: individuals with delirium; DEP, individuals with depressive symptoms; CONT, individuals without delirium or psychopathology. *d*, Cohen’s *d* (effect size: ≈0.2 small, ≈0.5 medium, and ≈0.8 large); PaO_2_, oxygen partial pressure, PaCO_2_, carbon dioxide partial pressure; SpO_2_, oxygen saturation; PaO_2_/FiO_2_ ratio of arterial oxygen partial pressure to fractional inspired oxygen. SD, standard deviation. * *p*-value switched to not significant after correction for age, education, comorbidities, pharmacological treatments before the admission, ADL and IADL before the admission. **^#^**
*p*-value became significant after adjustment for age, education, comorbidities, drug treatments prior to admission, ADL and IADL before admission.

**Table 3 jpm-13-01207-t003:** Significant predictors of respiratory and outcome variables in univariate/multivariate analyses in patients with depressive symptoms (DEP) (n = 42) and patients with delirium (DEL) (n = 42).

Group	Outcome	Predictors	Univariate Analyses					Multivariate Analyses				
			R^2^	β	*t*	95% CI	*p*	R^2^	β	*t*	95% CI	*p*
**DEP**	**PaO_2_**											
		HAM-D	0.12	−0.38	−2.55	−1.70, −0.19	**0.015**	0.34	−0.28	−2.03	−1.39, −0.002	**0.041**
		Dyspnea	0.14	−0.40	−2.71	−18.77, −2.72	**0.010**	0.34	−3.27	−2.41	−16.22, −1.41	**0.021**
		Drug treatments prior to admission	0.11	−0.37	−2.51	−3.46, −0.35	**0.017**	0.34	−2.87	−2.11	−1.39, −0.002	**0.049**
	**PaO_2_/FiO_2_**											
		HAM-D	0.17	−0.43	−2.99	−10.50, −2.02	**0.005**	0.32	−4.04	−2.80	−10.92, −0.26	**0.012**
		HAM-A	0.08	−0.32	−2.07	−7.92, −0.94	**0.045**	0.32	−0.171	−1.41	−5.31, 0.956	0.167
		Dyspnea	0.31	−0.57	−4.35	−130.76, −47.78	**<0.001**	0.32	−0.49	−4.37	−112.95, −41.44	**<0.001**
		Drug treatments prior to admission	0.15	−0.41	−2.82	−21.12, −3.51	**0.007**	0.32	−0.31	−2.72	−16.08 −2.36	**0.010**
	**Duration of hospitalization (days)**											
		BPRS	0.16	0.43	2.98	0.35, 1.83	**0.005**	0.52	0.31	2.78	0.22, 1.37	**0.008**
		Employed	0.08	−0.32	−2.13	−13.99, −0.35	**0.040**	0.52	−0.186	−1.67	−9.26, 0.88	0.103
		ADL	0.39	−0.63	−5.20	−5.07, −2.23	**<0.001**	0.49	−0.59	−5.22	−4.67, −2.06	**<0.001**
**DEL**	**PaO_2_**											
		NEECHAM	0.35	0.59	3.45	0.88, 3.53	**0.002**	0.62	0.56	4.11	1.02, 3.12	**0.001**
		Fever	0.28	−0.56	−3.15	−29.12, −6.01	**0.005**	0.62	−0.52	−3.83	−25.18, −7.46	**0.008**
	**PaO_2_/FiO_2_**											
		NEECHAM	0.34	0.61	3.58	5.62, 21.12	**0.002**	0.61	0.60	4.55	7.16, 19.31	**<0.001**
		Dyspnea	0.15	−0.44	−2.26	−156.88, −6.84	**0.034**	0.61	−0.27	−1.64	−116.47, 13.84	0.116
		Fever	0.26	−0.54	−2.99	−169.72, −30.76	**0.007**	0.61	−0.330	1.98	−125.92, 3.08	0.061

Note: Significant *p*-values (*p* < 0.05) are indicated in **bold**. DEL, patients with delirium; DEP, patients with depressive symptoms. PaO_2_, oxygen partial pressure, PaCO_2_, carbon dioxide partial pressure; SpO_2_, oxygen saturation; PaO_2_/FiO_2_ ratio of arterial oxygen partial pressure to fractional inspired oxygen. ADL, activities of daily living scale; BPRS, Brief Psychiatric Rating Scale; HAM-A, Hamilton Anxiety Rating Scale; HAM-D, Hamilton Depression Rating Scale, NEECHAM, Neelson and Champagne Confusion Scale. SD, standard deviation.

## Data Availability

Available in electronic form upon reasonable request to the corresponding author.

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
