# Peer review of "Association of Delirium and Depression with Respiratory and Outcome Measures in COVID-19 Inpatients"

_jpm, 2023, doi:10.3390/jpm13081207_

Round 1
Reviewer 1 Report (New Reviewer)
According to the article, the presence of delirium is associated with lower levels of respiratory parameters and a worse prognosis than the presence of depression in COVID-19 wards. In addition, severity of psychopathology and delirium predict lower levels of PaO2 and PaO2/FiO2. These findings suggest that resolution of neuropsychiatric syndromes could improve respiratory symptoms and help apply treatment strategies against COVID-19.
Unfortunately, the paper does not provide enough detailed information on data collection and analysis for this study. However, the article states that this is a retrospective observational study conducted on patients admitted to COVID-19 wards in an Italian hospital.
According to the article, the knowledge gained from this study could promote new efforts in early intervention strategies aimed at resolving neuropsychiatric comorbidity in people with COVID-19. Resolution of these neuropsychiatric syndromes could improve respiratory symptoms and, consequently, could help apply treatment strategies against COVID-19. In addition, further studies could clarify the impact of delirium and depression on the severity and course of COVID-19.
To me, it seems that the study has some limitations, such as a relatively small sample size and lack of longitudinal design. These limitations may affect the generalizability of results and limit the definition of a causal relationship between neuropsychiatric disorders and respiratory and outcome parameters. To address the limitations of the study, the authors could consider conducting a larger-scale study with a longitudinal design to clarify the dynamics between delirium/depression and respiratory and outcome variables. Additionally, they could include a non-COVID-19 control group to compare with and assess whether the differences found were specific to COVID-19. Additionally, the authors could also provide more detailed information on data collection and analysis methods to improve transparency and reproducibility. Finally, they could consider using more comprehensive evaluation methods for other variables involved in the relationship between neuropsychiatric symptoms and respiratory and outcome variables.
Overall, the language seems appropriate for an academic article in this field.
Author Response
In this response we kept your original observations and respond point-to-point just underneath each issue raised. Please note that changes in the manuscript will appear in red-coloured characters.
According to the article, the presence of delirium is associated with lower levels of respiratory parameters and a worse prognosis than the presence of depression in COVID-19 wards. In addition, severity of psychopathology and delirium predict lower levels of PaO2 and PaO2/FiO2. These findings suggest that resolution of neuropsychiatric syndromes could improve respiratory symptoms and help apply treatment strategies against COVID-19.
Unfortunately, the paper does not provide enough detailed information on data collection and analysis for this study. However, the article states that this is a retrospective observational study conducted on patients admitted to COVID-19 wards in an Italian hospital.
According to the article, the knowledge gained from this study could promote new efforts in early intervention strategies aimed at resolving neuropsychiatric comorbidity in people with COVID-19. Resolution of these neuropsychiatric syndromes could improve respiratory symptoms and, consequently, could help apply treatment strategies against COVID-19. In addition, further studies could clarify the impact of delirium and depression on the severity and course of COVID-19.
Thank you for the appreciation. You identified the gist of the study with great precision.
To me, it seems that the study has some limitations, such as a relatively small sample size and lack of longitudinal design. These limitations may affect the generalizability of results and limit the definition of a causal relationship between neuropsychiatric disorders and respiratory and outcome parameters. To address the limitations of the study, the authors could consider conducting a larger-scale study with a longitudinal design to clarify the dynamics between delirium/depression and respiratory and outcome variables. Additionally, they could include a non-COVID-19 control group to compare with and assess whether the differences found were specific to COVID-19. Additionally, the authors could also provide more detailed information on data collection and analysis methods to improve transparency and reproducibility. Finally, they could consider using more comprehensive evaluation methods for other variables involved in the relationship between neuropsychiatric symptoms and respiratory and outcome variables.
Thank you for these observations. We acknowledged all pitfalls in the Limitations section. We could not compare our sample to a non-COVID-19 one because it was not going to be a parallel group. We identified cases from our service only, which did not deal with non-COVID-19 cases. Retrospectively recruited groups would not be a suitable response. We thank you for useful suggestions that allowed us to improve our paper.

Reviewer 2 Report (New Reviewer)
In this interesting study, COVID-19 inpatients were divided in three groups according to the presence of delirium , depression (DEP) or absence of either delirium or depression (CONT). The authors considered the following respiratory parameters: partial pressure of oxygen in arterial blood (PaO2), oxygen saturation (SpO2), ratio between arterial partial pressure of oxygen (PaO2), and fraction of inspired oxygen (PaO2/FiO2). The outcome parameters were duration of hospitalization and number of pharmacological treatments used during the hospitalization. They found that the presence of delirium was associated with lower levels of respiratory parameters and poorer prognosis than the presence of depression in COVID-19 units. Furthermore, severity of depression and severity of delirium predicted lower levels of PaO2 and PaO2/FiO2. They concluded that mechanisms by which delirium and depression worsen respiratory parameters differ.
Questions:
1) The authors considered participants in the DEP group if the depression symptoms lasted more than two days. How was the rationale? Montgomery et al. (1989) described intermittent 3-day depressions and suicidal behavior. I wonder if this can be extrapolated to the HAM-D scale.
2) Regarding the HAM-D scale, previous investigators have considered two distinct dimensions. DSM-V criteria for depression have been analyzed in conjunction with the HAM-D scale and the same two dimensions of depression symptoms have emerged. The somatic dimension (moderate depression, symptoms using the HAM-D scale) has been found to be associated with a significant increase in QT dispersion. I wonder if the authors could perform an analysis of DEP effect on respiratory parameters based on these clusters, instead of simply considering the whole scale.
3) Confounders: I did not understand the rational for the selection of confounders. I would suggest performing a correlational analysis to help in the selection of confounders. Moreover, the selection of confounders based on statistical differences (e.g., age and educational level) is controversial. Even in matched designs the inclusion or exclusion of confounders needs to be carefully explained.
4) The number of pharmacological treatments used during the hospitalization was used as an outcome. What does it really mean? I suggest the description of all drugs that were used.
5) Although significance was reached, some values of the correlation coefficients (table 3) are small. How do the authors interpretate these small significant values.
6) What is the meaning of “d” in table 1?
7) I would suggest describing the effect size of the results.
Author Response
In this response we kept your original observations and respond to them just under each point raised. Our changes in the manuscript's text will be highlighted by red-coloured characters.
In this interesting study, COVID-19 inpatients were divided in three groups according to the presence of delirium , depression (DEP) or absence of either delirium or depression (CONT). The authors considered the following respiratory parameters: partial pressure of oxygen in arterial blood (PaO2), oxygen saturation (SpO2), ratio between arterial partial pressure of oxygen (PaO2), and fraction of inspired oxygen (PaO2/FiO2). The outcome parameters were duration of hospitalization and number of pharmacological treatments used during the hospitalization. They found that the presence of delirium was associated with lower levels of respiratory parameters and poorer prognosis than the presence of depression in COVID-19 units. Furthermore, severity of depression and severity of delirium predicted lower levels of PaO2 and PaO2/FiO2. They concluded that mechanisms by which delirium and depression worsen respiratory parameters differ.
Thank you for having appreciated our study and for finding interesting.
Questions:
1) The authors considered participants in the DEP group if the depression symptoms lasted more than two days. How was the rationale? Montgomery et al. (1989) described intermittent 3-day depressions and suicidal behavior. I wonder if this can be extrapolated to the HAM-D scale.
It was exactly on Stuart Montgomery’s criteria (Ref. 32) that we were based to establish our cutoff.
2) Regarding the HAM-D scale, previous investigators have considered two distinct dimensions. DSM-V criteria for depression have been analyzed in conjunction with the HAM-D scale and the same two dimensions of depression symptoms have emerged. The somatic dimension (moderate depression, symptoms using the HAM-D scale) has been found to be associated with a significant increase in QT dispersion. I wonder if the authors could perform an analysis of DEP effect on respiratory parameters based on these clusters, instead of simply considering the whole scale.
No, we did not and do not intend to endorse the somatic/psychic dimensions of the HAM-D and we explained why in the text (Limitations).
3) Confounders: I did not understand the rational for the selection of confounders. I would suggest performing a correlational analysis to help in the selection of confounders. Moreover, the selection of confounders based on statistical differences (e.g., age and educational level) is controversial. Even in matched designs the inclusion or exclusion of confounders needs to be carefully explained.
Thank you for the suggestion. We selected confounders by using regression, which looks at how one variable affects another, not only whether two independent variables are associated, as does correlation. We had already explained that in Statistical analysis.
4) The number of pharmacological treatments used during the hospitalization was used as an outcome. What does it really mean? I suggest the description of all drugs that were used.
It would be tedious to mention all drugs, which varied by much in the patient group. What we needed to say was that intake of more drugs was related to a worse health status, that would then affect respiratory parameters. This is why we chose to have the number of drug used as an outcome.
5) Although significance was reached, some values of the correlation coefficients (table 3) are small. How do the authors interpretate these small significant values.
Thank you for the observation; we added in Discussion that the stronger correlations were less likely to disappear on multivariate analyses.
6) What is the meaning of “d” in table 1?
Thank you for observing this; it was Cohen’s d, a measure of effect size.
7) I would suggest describing the effect size of the results.
Thank you for the suggestion; we described this through Cohen’s d coefficient in Tables. We also added in Table 1 the meaning of the Cohen’s d values, as generally accepted in literature. We thank you for your fine suggestions that helped our paper to improve by much.

Reviewer 3 Report (New Reviewer)

Author Response
Dear Reviewer,
In this response we kept your original observations and respond point-to-point just underneath each point raised. Please find our changes to the original version highlighted in red-coloured text.
Observations and Responses:
Congratulations on your work. Undoubtedly, it is a contribution that allows to deepen in the impact that psycho-emotional factors have on the clinical evolution of patients diagnosed with COVID-19.
There are some aspects, both general and specific, that should be reviewed.
Thank you for the generally positive attitude towards our manuscript.
General comments
- Throughout the document, check that bibliographic citation numbers are placed before punctuation marks.
I don’t know whether this is universal. Usually Europeans place them before punctuation, but Americans do the opposite. We preferred to adopt the American style, since it’s easier for readers accustomed to this style to follow the flow of the paper better. Since the publishers did not recommend us to anticipate citations, we left them as they were.
- Please remove the final periods from the titles of tables and figures.
We removed them.
- In the enumerations the components are separated by commas "," except for the last element of the enumeration which is preceded by "and" but without comma before "and". Please check throughout the document and modify.
Thank you, but as we previously stated, we used English American style throughout our manuscript. Americans require that in an enumeration, the final entity after the “and” is preceded by a comma. This is unnecessary for the British, but has some rationale for the Americans. Hence, we kept it as it was.
- Check that end-of-line hyphens separate complete syllables.
We can’t control for this; the system the Publishers adopt handles the entire manuscript and puts hyphens at its pleasure.
Page 1. Abstract
Authors should include the objective of the study.
Thank you for the suggestion, we included aim/objective.
Page 2. Keywords
“Partial arterial oxygen pres-sure (PaO2); Oxygen saturation (SpO2); Partial arterial oxygen pressure/inspired oxygen fraction ratio (PaO2/FiO2)” do not seem very appropriate terms to be included as keywords. Perhaps they could be included in Respiratory System parameters or some similar term.
Thank you for the tip. Partial pressure and saturation are often found among keywords. We omitted the Partial arterial oxygen pressure/inspired oxygen fraction ratio from keywords and added the generic term respiratory outcomes.
Page 2. Introduction
- "A surge or worsening of several neuropsychiatric disorders, such as delirium, dementia, depression, anxiety, phobias, and manic states, was observed in these units. Among all the aforementioned neuropsychiatric disorders, delirium and depression were the most frequently observed." It should be clarified that these neuropsychiatric disorders are not exclusive to the specific units for patients with -COVID-19 but also have a high incidence in people admitted to Intensive Care Units, regardless of the pathology they present.
Thank you for the suggestion. We clarified.
- "The onset of delirium or depression inpatients with COVID-19 might affect their prognosis. People in COVID-19 units who developed delirium showed PaO2 and PO2/FiO2 levels shifted toward the pathological side, greater duration of hospitalization, greater rates of in-hospital death, greater use of intensive ventilators, and higher admission rates to intensive care" Along the same lines as the previous comment, these aspects are not specific to patients with COVID-19, they are common to any other respiratory pathology.
Thank you for the prompt, we specified that.
- "Since hospitalized patients with delirium have shown a poorer prognosis than those affected by depression or those without either delirium or de-pression, we expected to find lower PaO2, SpO2, PaO2/FiO2, and higher PaCO2 in DEL as compared with DEP and CONT. To investigate the association of severity of delirium/de-pression with the levels of respiratory parameters and prognosis, we used psychometric scales usually employed to assess delirium, depression, and other psychopathology associ-ated with depression, such as anxiety, manic and psychotic symptoms, irritability, and ap-athy." This sentence corresponds more to the "Methodology" section than to the "Introduction" section; it is not correctly placed between the objectives and the hypothesis of the study.
In Methods we specified better, in Introduction we introduced the rationale for the study.
Page 4. Sample
- How did you calculate the optimal sample size?
We used GraphPad StatMate, but now we see it’s no longer available. We turned to ClinCalc which provided the figures we describe.
- "Informed consent was obtained by participants before entering the study".
- Did the clinical and physiological condition of the participants allow, in all cases, that they personally signed their informed consent?
Yes, otherwise we did not include the patient.
- Unifying the sample eligibility criteria in the same paragraph would make it easier to locate them. The inclusion criteria are located in one place, in Figure 1, and some exclusion criteria are located at the end of the "sample" subsection, which makes it difficult to consult them.
Thank you for noticing. Actually, both inclusion and exclusion criteria are shown in Figure 1, but those exclusionary criteria you found at the end of the section regard patients who were eligible in the first instance, but had to be excluded for one of the reasons specified at the end of the Sample subsection. If we anticipate them, we would interrupt the logical flow of the manuscript.
Page 5. Assessment tools
In which language were the scales used in this study developed, and was any translation/cultural adaptation/validation of these scales into Italian performed before using them in this research? Please, specify this aspect in the methodology
The original language was English for all scales; we added a sentence to state we used validated, Italian versions of each scale. All scales are used since decades in Italian research.
Page 6. Respiratory parameters
- Why was it decided to measure the respiratory parameters within 12 hours after the psychopathological assessment? Was it not performed, in all participants, at the same time after the psychopathological assessment?
It could vary among patients. We added in Limitations.
- "Gas analysis performed by a physician" Was it always the same physician? Was the physician blinded?
Physicians were those at shift, quite expert and consistent in their results. They were not blinded as to the patient’s conditions, but were blind as to the aims of the research and to the fact that the patient was a participant in our research. We added details.
- The methodology for obtaining the arterial blood sample, the type of pulse oximeter used and its range of error in measurement, whether the arterial blood sample analysis was performed in a laboratory or with a portable analyzer, … need to be further specified.
We specified each of these.
- The methodology should allow any other investigator to replicate the study.
Page 6. Other scales related to the clinical status
Please elaborate further on the scales included in this section.
Any other investigator needing to perform a similar study will find enough material to “replicate” the study, but nobody will be able to truly replicate it, since COVID-19 times are passing and the conditions that engendered the study are no more reproducible. Further elaborating on scales that matter so little to this study would unnecessarily expand the text without adding any useful information for researchers needing to do a remake. They could consult the References list.
Page 6. Ethics
Please provide the identification number of the favorable report from the Ethics Committee.
We provided the ID in both text and ethical statement.
Page 9. Table 2
In Table 2 there are p-values less than 0.05 that are not marked in bold.
Thank you for observing. We turned them into bold.
Page 13. Discussion
"Absence of direct comparisons between pa-tients with delirium and depression in COVID-19 inpatients dampens the formulation of possible hypotheses on why DEL showed greater duration hospitalization than DEP" And what do studies in patients with respiratory failure other than COVID-19 say? Do they agree or differ with these results?
Thank you for the observation. In non-COVID-19 contexts delirium or depression do not affect hospitalization rate. We added a referenced sentence to make it clearer.
Supplementary files
Sup. table 2 and sup. table 3.
There are p-values less than 0.05 that are not highlighted in bold.
Thank you for this observation; we corrected the supplement. We thank you for your thoughtful suggestions that improved by much our manuscript.

This manuscript is a resubmission of an earlier submission. The following is a list of the peer review reports and author responses from that submission.
Round 1
Reviewer 1 Report
Age of the patients might play a role in the development of delirium and depression. This should be mentioned in the discussion and or limitations.
The DEL group had a higher number of drugs also before hospital admission. As the number of medication might influence the occurance of POD this should be discussed.
Author Response
We carried out the revision asked by your two reviewers. We highlighted changes in the text in red-coloured characters, so to facilitate comparison with the previous version. We hope that you and your reviewers will find this new version to be suitable for publication in the Journal of Personalized Medicine. To help Reviewers to check for changes made upon their suggestions, we keep in this response their original observations and respond underneath each point raised.
Reviewer 1
Age of the patients might play a role in the development of delirium and depression. This should be mentioned in the discussion and or limitations.
We thank Reviewer #1 for this suggestion. We mentioned this in Discussion and Limitations. The added sentences are shown below:
Discussion:
Another important factor that could affect both the expression of delirium and the occurrence of depression is advancing age. In fact, it has been shown that age is the most important factor determining the occurrence of delirium in surgery wards (Kubota et al., 2018), and it could be so in ICUs (Gao et al., 2021), where all too often patients with severe COVID-19 are referred (La Hue et al., 2022); in all cases, higher age increases the odds of people developing delirium and worsens outcomes. Although late-life depression has its own clinical peculiarities (Alexopoulos, 2019), the relationship of the onset of depression across ages is still developing and may be complex (Beller, 2022). It appears that delirium and depression share common pathophysiological mechanisms and can trigger one another (O’Sullivan et al., 2014). Their interrelations with age deserve to be better studied in future work.
Limitations:
We did not stratify our sample according to age, that is a definite and important factor associated with both delirium and depression, but we entered age as a possible con-founder in our analyses of variance and the subsequent regressions.
The DEL group had a higher number of drugs also before hospital admission. As the number of medication might influence the occurance of POD this should be discussed.
We discussed the possible influence of the number of medications on POD occurrence in the DEL group. We thank Reviewer #1 for the useful suggestions he/she provided that helped us producing a better manuscript. Below you can find the text added:
In our study, patients with delirium were taking a higher number of medications on admission. This is not unexpected, as the number of medications proved to be predictive of future delirium (Gao et al., 2008; Lahariya et al., 2014; Levinoff et al., 2018; Zhu et al., 2020), but also patients with depression are likely to be treated with anticholinergic drugs, that may induce confusion and delirium (Kassie et al., 2019). However, in our sample, the depressive group did not differ from nondepressive-nondelirium controls.

Reviewer 2 Report
The authors conducted a study on patients infected with the SARS-CoV-2 virus admitted to a hospital. The patients were divided on groups with delirium, depression or the absence of both. The primary objective was to investigate differences in respiratory and outcome parameters of inpatients with COVID-19 between those having delirium or those with depression or those without delirium or depression; ii) to investigate if severity of delirium or depression is differentially related with respiratory and outcome parameters in the above two groups.
There are many serious flaws in the study. To name the most important:
- The authors classified their study as a cross-sectional study when in reality it is a retrospective analysis, it is unclear if they collected the information prospectively or if they collected the information retrospectively.
-There was no sample size calculation, making the results difficult to interpret in this setting
- The authors used an old confusion score to classify the patients with delirium, when there is an extensively validated and contemporary tool like the CAM.
- They failed to mention the single most important delirium characteristic: its fluctuating state, which differentiate it from dementia.
- They included the diagnosis of depression; however, there is no mention of temporality, meaning if the patient had previously been diagnosed with depression. Temporality is important, because the DSM-5 for depression requires 2 weeks of symptoms, the authors included patients after 1 week of hospitalization, making It impossible to diagnose acute depression, specially since they claimed there is a correlation with respiratory outcomes
- There was not a plausible physiological justification to perform the study; nor there was a scientific elaboration in the discussion of their findings
- The introduction is too long and contains information that should be in the methods.
- The algorithm presented in the methods is confusing and unnecessary because they elaborated in text anyway.
- As I mentioned before the type of study claimed does not correlate with the information presented.
- It is not clear if the patients were included consecutively or which method they used to screen them.
- The inclusion/exclusion criteria are not clearly defined. There is some information in the introduction and another in the methods.
- In the paragraph of the effect of possible confounding variables, the authors mentioned that they used ANOVA, and that is confusing given that it is not use for that purpose (in any case it should be ANCOVA)
- There was an intent to control for confounding factors; however, they failed to capture severity scores that potentially influence those outcomes like SOFA or APACHE
- The results showed some statistically difference between groups in the respiratory outcomes; however, the differences, although statistically significant, do not have any clinical value, nor would change management.
- The only significant difference that make sense is the observed increased hospital length of stay among patients with delirium, observation that was previously and extensively demonstrated.
- The document requires extensive review of the English grammar and style.
Author Response
The authors conducted a study on patients infected with the SARS-CoV-2 virus admitted to a hospital. The patients were divided on groups with delirium, depression or the absence of both. The primary objective was to investigate differences in respiratory and outcome parameters of inpatients with COVID-19 between those having delirium or those with depression or those without delirium or depression; ii) to investigate if severity of delirium or depression is differentially related with respiratory and outcome parameters in the above two groups.
There are many serious flaws in the study. To name the most important:
- The authors classified their study as a cross-sectional study when in reality it is a retrospective analysis, it is unclear if they collected the information prospectively or if they collected the information retrospectively.
We thank Reviewer 2 for this observation. As mentioned by Reviewer 2, some aspects of the methodology used appear to relate to a retrospective design. Confusion in the methodology might have risen from the exclusion of subjects with delirium further developing depression and vice-versa. However, longitudinal aspects of the methodology are limited to the inclusion/exclusion criteria. The present study subjects din not receive more than one assessment, and the analyses were not performed longitudinally. Therefore, we prefer to maintain the term “cross-sectional”.
-There was no sample size calculation, making the results difficult to interpret in this setting
We thank Reviewer 2 for this observation. We actually performed sample size calculation. We recruited a population of 100 patients and the sample size calculator stated that at least 80 patients were needed for a 95% confidence level and a 5% margin of error. Hence, the results are not difficult to interpret.
- The authors used an old confusion score to classify the patients with delirium, when there is an extensively validated and contemporary tool like the CAM.
We thank Reviewer 2 for this observation. We used the Neelon and Champagne scale (NEECHAM) of 1996, which is frequently used in recent times, with 15 occurrences on PubMed from 2017 to 2022 and provides a clear cut-off for delirium. The Confusion Assessment Method (CAM), which is not younger than the NEECHAM (in fact, it was developed six years before, Inouye SK, van Dyck CH, Alessi CA, Balkin S, Siegal AP, Horwitz RI. Clarifying confusion: the confusion assessment method. A new method for detection of delirium. Ann Intern Med. 1990 Dec 15;113(12):941-8. doi: 10.7326/0003-4819-113-12-941. PMID: 2240918), is in fact much more used, but its use largely focuses on Intensive Care Units (ICUs); you should recall that our study excluded patients transferred to ICUs. However, the NEECHAM has shown better sensitivity than the CAM (Van Rompaey B, Schuurmans MJ, Shortridge-Baggett LM, Truijen S, Elseviers M, Bossaert L. A comparison of the CAM-ICU and the NEECHAM Confusion Scale in intensive care delirium assessment: an observational study in non-intubated patients. Crit Care. 2008;12(1):R16. doi: 10.1186/cc6790. Epub 2008 Feb 18. PMID: 18282269; PMCID: PMC2374628), so our choice seems more than justified. We added this in the Methods section to clarify the issue from the beginning:
We chose the NEECHAM instead of the Confusion Assessment Method (CAM) (Inouye et al., 1991) to evaluate delirium because the former showed better sensitivity in detecting delirium cases (Van Rompaey et al., 2008).
- They failed to mention the single most important delirium characteristic: its fluctuating state, which differentiate it from dementia.
We thank Reviewer 2 for this most important observation. We mentioned it in the text:
Delirium can be differentiated from dementia due to its fluctuating nature, but in cases in which it superimposes dementia, the differential diagnostic algorithm becomes more complicated (Ciampi et al., 2011).
- They included the diagnosis of depression; however, there is no mention of temporality, meaning if the patient had previously been diagnosed with depression. Temporality is important, because the DSM-5 for depression requires 2 weeks of symptoms, the authors included patients after 1 week of hospitalization, making It impossible to diagnose acute depression, specially since they claimed there is a correlation with respiratory outcomes
We thank Reviewer 2 for observing this. There is a mention of temporality in the description of the groups, where we stated that “participants with depressive symptoms lasting less than the two minimum days required to define a brief depression (Montgomery et al., 1989)” were to be excluded. The DSM-5, according to itself, requires “Five (or more) of the following symptoms [Depressed most of the day, nearly every day as indicated by subjective report (e.g., feels sad, empty, hopeless) or observation made by others (e.g., appears tearful); Markedly diminished interest or pleasure in all, or almost all, activities most of the day, nearly every day (as indicated by subjective account or observation); Significant weight loss when not dieting or weight gain (e.g., change of more than 5% of body weight in a month), or decrease or increase in appetite nearly every day; Insomnia or hypersomnia nearly every day; Psychomotor agitation or retardation nearly every day (observable by others, not merely subjective feelings of restlessness or being slowed down); Fatigue or loss of energy nearly every day; Feelings of worthlessness or excessive or inappropriate guilt (which may be delusional) nearly every day (not merely self-reproach or guilt about being sick); Diminished ability to think or concentrate, or indecisiveness, nearly every day (either by subjective account or as observed by others); Recurrent thoughts of death (not just fear of dying), recurrent suicidal ideation without a specific plan, or a suicide attempt or a specific plan for committing suicide] have been present during the same 2-week period and represent a change from previous functioning; at least one of the symptoms is either (1) depressed mood or (2) loss of interest or pleasure” to be present for a major depressive episode diagnosis to be posed. However, in our paper, we never mentioned the DSM-5 nor diagnoses of major depressive disorder or episodes. In fact, the group “DEP” was defined as a group having a HAM-D score >7. In many works, this threshold has used to define presence of depression in spite of DSM-5 criteria. Additionally, in order to be more cautious, we defined as “DEP” those having depressive symptoms without mentioning major depressive disorder or episode.
- There was not a plausible physiological justification to perform the study; nor there was a scientific elaboration in the discussion of their findings
We thank Reviewer 2 for this observation.
- The introduction is too long and contains information that should be in the methods.
Reviewer’s 2 opinion is respectable. We fail to understand which part of the Introduction section should be moved to methods, as Reviewer 2 did not mention anything specific.
- The algorithm presented in the methods is confusing and unnecessary because they elaborated in text anyway.
We thank Reviewer 2. As pointed out by Reviewer 2, methodology related to this study is quite complex. We would prefer to keep the algorithm in order to ease the readability of the study design.
- As I mentioned before the type of study claimed does not correlate with the information presented.
We thank Reviewer 2 for returning to his standpoint that our study was not cross-sectional. We believe we settled this point above.
- It is not clear if the patients were included consecutively or which method they used to screen them.
We thank Reviewer 2. The method is extensively described in Methods. Under the Sample subheading, we wrote “Inpatients enrolled were consecutively hospitalized for SARS-CoV2 infection”.
- The inclusion/exclusion criteria are not clearly defined. There is some information in the introduction and another in the methods.
We accept Reviewer’s 2 comments. Nevertheless, information found in the Introduction is sound and logical by our opinion, since you cannot introduce an issue without stating something about it and without providing definitions. Inclusion and exclusion criteria are all found in detail in the Methods and also in Figure 1.
- In the paragraph of the effect of possible confounding variables, the authors mentioned that they used ANOVA, and that is confusing given that it is not use for that purpose (in any case it should be ANCOVA)
We thank Reviewer 2 for this observation. We did not understand Reviewer’s 2 point of view. We try to clarify our methodology: In order to investigate the effect of possible confounding variables on each ANOVA, all the ANOVAs were re-run with possible confounding variables in the model. In agreement with Reviewer 2, ANOVA were substituted by ANCOVA. We rephrased the sentence regarding this passage in the appropriate section:
Therefore, multiple one-way analyses of covariance (ANCOVAs) were run. In each ANCOVA, the three groups (DEL, DEP, CONT) were used as independent variables; continuous variables of interest were used as dependent variables and possible con-founding variables were used as covariates.
- There was an intent to control for confounding factors; however, they failed to capture severity scores that potentially influence those outcomes like SOFA or APACHE
We accept Reviewer’s 2 comment. However, the Sequential Organ Failure Assessment (SOFA) score is used to predict mortality in ICUs as well as the Acute Physiology and Chronic Health Evaluation II (APACHE-II). In our study, admission in the ICU is an exclusion criteria. Therefore we do not consider these two scales as appropriate.
- The results showed some statistically difference between groups in the respiratory outcomes; however, the differences, although statistically significant, do not have any clinical value, nor would change management.
We thank Reviewer #2 for this observation. However, we believe that when you deal with a new illness entity which has no definite treatment plan to apply, it is surely important to gather data that you will be able to use in a near future, as management requirements are newly discovered and treatment plans change.
- The only significant difference that make sense is the observed increased hospital length of stay among patients with delirium, observation that was previously and extensively demonstrated.
We thank Reviewer 2. We agree with Reviewer 2 that the finding of a greater length of hospitalization in subjects with delirium as been already demonstrated. However, we believe in the significance of other findings present in our study
- The document requires extensive review of the English grammar and style.
We thank Reviewer 2 for this comment. However, the English has been already written, revised and double checked by two mother tongue English speakers.

Round 2
Reviewer 2 Report
1. The aim of this study was twofold: i) to investigate differences in respiratory and outcome parameters of inpatients with COVID-19 between those having delirium or those with depression or those without delirium or depression; ii) to investigate if severity of delirium or depression is differentially related with respiratory and outcome parameters in the above two groups.
Their results showed statistical differences between groups; however, there is absolutely not clinical significance of a PF ratio difference of 30-40 points when all are above 300 (normal!) or a SpO2 of 2 when all values are above 93% (normal!). The lack of clinical significance of this differences does not support their conclusion. Furthermore, the authors define their study as cross-sectional, therefore, we have not enough information to determine if those values were present before the delirium or the diagnosis of depression.
2. On Table 3 Significant predictors of respiratory and outcome variables in univariate/multivariate analyses in subjects with depressive symptoms (DEP) and subjects with delirium (DEL), the authors presented different R2 results, again, poor clinical correlations.
The big issue here is that we do not have enough data to draw strong conclusions, there are so many uncontrolled variables like age, the severity of covid, the presence/abscence of COPD or previous respiratory diseases. The results are not clinical relevant.